# Validation of the Chinese Version of the Body Image Flexibility and Inflexibility Scale among Chinese College Students

**DOI:** 10.3390/bs14100910

**Published:** 2024-10-08

**Authors:** Ruichao Jiao, Dan Zheng, Dongdong Xue, Xiaowei Guo, Hongxing Meng, Xiaozhuang Wang

**Affiliations:** 1Faculty of Psychology, Tianjin Normal University, Tianjin 300387, China; jrc6767@hueb.edu.cn (R.J.); zhengdan@nnnu.edu.cn (D.Z.);; 2Student Affairs Office, Hebei University of Economics and Business, Shijiazhuang 050061, China; 3College of Chemistry and Materials, Nanning Normal University, Nanning 530100, China; 4School of Educational Sciences, Xinxiang University, Xinxiang 453003, China; 5School of Psychology, Central China Normal University, Wuhan 430079, China; 6Key Research Base of Humanities and Social Sciences of the Ministry of Education, Academy of Psychology and Behavior, Tianjin Normal University, Tianjin 300387, China; 7Tianjin Social Science Laboratory of Students’ Mental Development and Learning, Tianjin 300387, China

**Keywords:** body image flexibility, body image inflexibility, body image flexibility and inflexibility scale, validation, college students

## Abstract

Body image flexibility is a specific type of psychological flexibility relevant to body image. The development of the Body Image Flexibility and Inflexibility Scale (BIFIS) expands the concept and structure of body image flexibility and provides more detailed measurement indicators for theoretical research and clinical practice. However, the tool’s applicability to the Chinese population is still unclear. This study aims to test the reliability and validity of the BIFIS among Chinese college students. A total of 1446 Chinese college students were surveyed and completed a series of scales, including the Chinese version of the BIFIS (i.e., C-BIFIS). A total of 99 participants were retested one month later. Confirmatory factor analysis supported the second-order factor structure of the BIFIS. The C-BIFIS showed measurement invariance across genders. The scale also exhibited good internal consistency and test–retest reliability. The higher-order body image flexibility and inflexibility factors were significantly correlated with unidimensional body image flexibility, body satisfaction, body appreciation, intuitive eating, and life satisfaction. Incremental validity tests indicated that two higher-order factors remained unique predictors of intuitive eating and life satisfaction. In conclusion, the Chinese version of the BIFIS has good psychometric properties and could be used to study body image flexibility in Chinese college student populations.

## 1. Introduction

Body image flexibility denotes a person’s capacity to voluntarily accept their body’s present experience (e.g., cognitions, emotions, bodily sensations) without judgment while pursuing chosen values [1,2]. As a specific form of psychological flexibility relevant to body image [3], body image flexibility encompasses all features of psychological flexibility [4]. Furthermore, it is considered a positive body image component [3,5]. Body image inflexibility is the opposite: attempts to change or delay aspects of the body experience that ultimately disrupt valued action [6]. Numerous studies indicate that body image flexibility is positively linked to various aspects of body image and physical and mental health. [2,3,6]. For example, body image flexibility has a negative correlation with negative psychological variables such as body dissatisfaction [7], eating disorders [8], unhealthy exercise behaviors [9], and motivation [10], as well as psychopathological symptoms [11]. On the other hand, it has a positive correlation with positive psychological variables, including body appreciation [12,13,14], intuitive eating [15,16], and life satisfaction [11]. Furthermore, body image flexibility is a positive factor in preventing disordered eating [17]. In clinical practice, it contributes to the effectiveness of third-wave behavior therapy interventions, represented by Acceptance and Commitment Therapy (ACT). Improving body image flexibility is often accompanied by success in eating disorder treatment [6]. In short, body image flexibility positively impacts individuals’ ability to establish a positive self-image assessment and promote their physical and mental health.

The current construct and measures of body image flexibility are derived from psychological flexibility, an individual’s capacity to consciously and fully experience the present moment and adhere to or change behavior by their chosen values, which is ACT’s core concept and the primary treatment goal [18]. Psychological flexibility is characterized by six core components: experiential acceptance, cognitive defusion, awareness of the present moment, self-as-context, valuing, and committed action. Conversely, psychological inflexibility is marked by its opposing dimensions: experiential avoidance, cognitive fusion, disconnection from the current moment, self-as-content, lack of contact with values, and inaction [18]. Body image flexibility, a specialized manifestation of psychological flexibility, was initially conceptualized as a singular, one-dimensional construct [1]. The most frequently used assessment tool is the Body Image-Acceptance and Action Questionnaire (BI-AAQ) [1]. The BI-AAQ is a unidimensional scale that has been adapted from the Acceptance and Action Questionnaire (AAQ) [19] and its revised version (AAQ-II) [20], both of which evaluate the broader concept of psychological flexibility. Subsequently, the BI-AAQ-5, an abbreviated form of the BI-AAQ, was created by Basarkod et al. [21] using genetic algorithms. Both the comprehensive and shortened versions of the BI-AAQ have exhibited robust psychometric properties when applied to assess clinical and non-clinical samples of adults [11,22,23]. Moreover, translations of the English-language BI-AAQ have been completed for a multitude of languages, such as Mandarin Chinese [24], Greek [25], Portuguese [26], and Spanish [27]. The development and use of the BI-AAQ have significantly advanced research in body image flexibility.

Nevertheless, the BI-AAQ presents certain constraints. Firstly, every item on the BI-AAQ was scored in reverse, and the incongruence between the negative wording of the items and the positive concept of body image flexibility has led to questions about its content validity [3]. Secondly, the measure’s content focuses only on body weight and size, which does not cover all aspects of body image. Thirdly, the BI-AAQ primarily concerns experiential avoidance (the opposite of acceptance) without considering the other core psychological flexibility processes. Some studies even call the BI-AAQ “body image avoidance” [28] and use it as a measurement of body image inflexibility [8,14,29,30,31,32]. To this end, several studies have emphasized that body image flexibility should be considered a distinct and multifaceted construct [2,3,5]. In a recent survey, Sandoz et al. [6] put forward a theoretical framework for assessing body image flexibility using six dimensions of psychological flexibility. Furthermore, Brichacek et al. [33] have crafted a comprehensive scale termed the Body Image Flexibility and Inflexibility Scale (BIFIS). This instrument is an adaptation of the abbreviated version of the Multidimensional Psychological Flexibility Inventor (MPFI-24) [34].

The BIFIS initially consisted of a 36-item pool designed for young people based on the MPFI-24. The expected factor structure is a second-order factor model with 2 higher-order and 12 lower-order factors, aligning with the structural composition of the MPFI-24. However, due to the correlation of some of the subfactors, the final construct was a second-order factor model with two higher-order and eight lower-order factors. The BIFIS consists of 24 items, with each of the eight subordinate factors encompassing 3 items. The four subfactors of higher-order body image flexibility are mindful acceptance (combed by acceptance and present-moment attention), cognitive defusion, self-as-context, and values connection (combed by values and committed action). In contrast, the four subfactors of higher-order body image inflexibility are active avoidance (combed by avoidance and lack of awareness), cognitive fusion, self-as-content, and values disconnection (combed by lack of values and inaction). The BIFIS is reliable and valid among young Australians and is measurement invariant across gender and age groups [34]. In contrast to the unidimensional BI-AAQ, the BIFIS focuses on the experience of the body as a whole rather than being limited to one component of the body (e.g., body size and weight). The BIFIS not only differentiates between body image flexibility and inflexibility but also refines their components, further extending the existing concept and structure of body image flexibility. Nonetheless, the scale’s psychometric attributes in other cultural groups remain unclear. The development or validation of instruments that can effectively measure the positive components of body image flexibility will form the basis for the cross-cultural study of body image flexibility [2]. Therefore, the cultural applicability of the BIFIS needs to be further tested.

Studies have investigated the beneficial impacts of body image flexibility on young Chinese people’s physical and psychological well-being. For example, Shi et al. [35] discovered that increased body image flexibility among Chinese medical students was associated with decreased intermittent fasting behavior. Meanwhile, body image flexibility was also a mediating variable in the association between young adults’ body weight [31], level of celebrity admiration [36], and emotional intelligence [8] with disordered eating. In other words, enhancing body image flexibility helps alleviate the negative effects of eating disorder risk factors. These findings provide empirical support for further cross-cultural study of body image flexibility. Nevertheless, the instruments used in the above studies were unidimensional BI-AAQ, which mainly assessed weight and body size. The physical self of Chinese youth includes height and appearance in addition to fatness and thinness [37]. Hence, a tool to assess body image flexibility that better reflects the whole body image picture is needed. In addition, most of the aforementioned research has focused on the dynamics of body image flexibility predominantly through the lens of its absence. As a favorable psychological construct, body image flexibility differs in structure and components from body image inflexibility [33]. Nevertheless, the collective influence of body image flexibility and inflexibility, along with the discrete roles of their various components, on the coping mechanisms employed by Chinese young adults in response to body image threats remains largely obscure. Exploring a multifaceted structure of body image flexibility and inflexibility in the Chinese cultural context is essential for uncovering the above mechanisms better.

Considering that the BIFIS has not yet been tested for applicability in a Chinese population, the primary aim of this investigation was to test the measurement characteristics of the BIFIS within a sample of Chinese university students. The specific aims of this research are as follows: (1) to translate the BIFIS into Chinese and to assess its reliability and validity among Chinese college students, and to confirm the scale’s second-order factor structure within the cultural milieu of China utilizing confirmatory factor analysis; (2) to further examine the cross-gender measurement invariance of the C-BIFIS to determine whether the scale is influenced by instrumental bias; (3) to contrast the associations between the C-BIFIS and existing unidimensional measures of body image flexibility; and (4) to examine the convergent and incremental validities of the C-BIFIS, i.e., the correlations of the C-BIFIS with the variables of body satisfaction, body appreciation, intuitive eating, and life satisfaction, as well as the unique predictive roles of two higher-order factors on intuitive eating and life satisfaction after controlling for related variables such as unidimensional body image flexibility.

## 2. Methods

### 2.1. Participants and Procedures

Using convenience sampling, 1600 students were selected to participate in the online survey at four universities in Hebei, Henan, and Guangxi provinces, China. Only full-time undergraduate students would be included. All participants met the condition. There were no missing data here as all questions were mandatory in the online survey. In order to prevent the participants from not answering carefully, the online survey set up an “assigned answer question”. Assigned response errors, regular responses (e.g., the same choice for all questions), and short response times (e.g., less than 200 s) were considered invalid questionnaires. According to these exclusion criteria, 154 invalid questionnaires were excluded, and 1446 valid participants were obtained (394 males, 1052 females; 857 first-year students, 264 sophomores, 245 juniors, 80 seniors; 480 from cities, and 966 from villages). The participants’ fields of study included engineering (390 students, 26.97%), science (328 students, 22.68%), management (293 students, 20.26%), economics (255 students, 17.64%), education (52 students, 3.60%), law (50 students, 3.46%), and other disciplines (78 students, 5.39%). Participants were aged 16 to 24 years (mean age 19.20 ± 1.39 years). The participant’s Body Mass Index (BMI) was calculated based on their reported weights and heights, ranging from 13.21 to 41.40 kg/m^2^ (mean BMI 21.05 ± 3.45 kg/m^2^). One month later, 130 participants were retested, and 99 matched and validated participants (15 males, 84 females; mean age 18.50 ± 0.75 years).

This research received clearance from the Scientific Research Ethics Committee of Tianjin Normal University. Before the assessment, counselors or psychological health teachers briefed the participants on the research’s objectives and procedures, ensuring that all engagement was based on voluntariness, devoid of any form of compensation or course credit. All participants completed an online informed consent form. The entire online survey took about ten minutes to complete.

### 2.2. Translation and Adaptation of the BIFIS

The right to translate and revise the original BIFIS was obtained with the author’s permission. Firstly, two postgraduate psychology students fluent in Chinese and English independently translated the English scale into Chinese. The initial Chinese version was produced after the first and second authors discussed and resolved the discrepancies between the two versions. Secondly, two Ph.D. students with backgrounds in bilingual learning of English and psychology back-translated the first Chinese version into English. An English professor with experience translating psychological works then compared the differences between the translated and back-translated versions and adjusted some expressions of the initial Chinese version. Finally, after evaluation by the corresponding author, a professor of psychology, the final Chinese version was produced. Appendix A provides the Chinese translation of the BIFIS (see Appendix A).

Prior to the formal administration of the test, a pre-survey study was conducted. Using a convenience sampling method, some first-year students were selected to read the content of the scale and asked whether the content of the items was clear and whether they could understand the meaning of the sentences. They reported that the items were clear and that they understood the meaning of the sentences. Therefore, no adjustments were made to the scale content.

### 2.3. Measures

#### 2.3.1. Chinese Version of the Body Image Flexibility and Inflexibility Scale (C-BIFIS)

Consistent with the original scale, the C-BIFIS consisted of 24 items to evaluate an individual’s ability to cope with threats about physical appearance [33]. The scale comprised two components: body image flexibility and inflexibility subscales. Each contained four dimensions and 12 items. The composite scores for the flexibility items (e.g., “I was aware of my thoughts and feelings about my body”) and the inflexibility items (e.g., “I tried to ignore my thoughts and feelings about my body”) were averaged to yield the respective scores for body image flexibility and inflexibility [38]. Participants were asked to rate each statement on a scale from 1 (strongly disagree) to 6 (strongly agree), where higher scores reflected a higher degree of either body image flexibility or inflexibility. Acknowledging that the temporal frame of the BIFIS instructions may be modified to suit specific requirements, Brichacek et al. [33] proposed this adaptability. Consequently, the Chinese rendition had adjusted the temporal reference from “the past four weeks” to “the previous month” for a better cultural fit.

#### 2.3.2. Unidimensional Body Image Flexibility

The abbreviated form of the Body Image-Acceptance and Action Questionnaire (BI-AAQ-5), a unidimensional measure, was employed to evaluate this construct. Originating with Sandoz et al. [1], the scale was subsequently streamlined by Basarkod et al. [21] to a short version with five items (e.g., “Feeling fat causes problems in my life”). He et al. [24] revised the BI-AAQ-5 into Chinese. Scoring for each item was based on a 7-point Likert scale ranging from 1 (never true) to 7 (always true), with items scored reversely to reflect that a higher composite score denoted enhanced body image flexibility [24]. The McDonald’s omega coefficient of this research was 0.85 [CIs 95% 0.84; 0.86].

#### 2.3.3. Body Satisfaction

The Body Areas Satisfaction Scale (BASS) was utilized to quantify the satisfaction with various body areas. This subscale originated from the Multidimensional Body-Self Relations Questionnaire crafted by Brown et al. [39] and had been culturally adapted into Chinese by the efforts of Wang and Wang [40]. The BASS encompassed 9 items (e.g., “Weight”) rated from 1 (very dissatisfied) to 5 (very satisfied). An elevated average score on this scale signified a higher level of overall body satisfaction. The McDonald’s omega coefficient of this research was 0.91 [CIs 95% 0.90; 0.92].

#### 2.3.4. Body Appreciation

The Body Appreciation Scale-2 (BAS-2) was adopted to assess an individual’s acceptance, positive sentiments, and respect for their body. This scale was initially developed by Tylka and Wood-Barcalow [41] and later adapted into a Mandarin Chinese version by Ma et al. [42]. The scale was composed of 10 items (e.g., “I respect my body”), and respondents rated these statements on a 5-point Likert scale from 1 (never true) to 5 (always true). An increased average score on the scale indicated greater body appreciation [33]. The McDonald’s omega coefficient of this study was 0.94 [CIs 95% 0.93; 0.94].

#### 2.3.5. Intuitive Eating

The Intuitive Eating Scale-2 (IES-2) was used to evaluate the degree to which a person relies on physiological mechanisms within the body to guide eating. The scale was first created by Tylka et al. [43] and culturally adapted into Chinese by Ma et al. [44]. It comprised 23 items (e.g., “I trust my body to tell me when to eat”) and was rated on a 5-point Likert scale from 1 (strongly disagree) to 5 (strongly agree). Certain items, including 1, 2, 4, 5, 9, 10, and 11, were scored reversely, where higher aggregate scores indicated more intuitive eating behavior [44]. The McDonald’s omega coefficient of this study was 0.69 [CIs 95% 0.67; 0.72].

#### 2.3.6. Life Satisfaction

The Satisfaction with Life Scale (SWLS) was used to measure an individual’s overall contentment with life. Diener et al. [45] developed the scale, which Xiong and Xu [46] later revised into Chinese. It comprised five items (e.g., “I am satisfied with my life”), which participants evaluated on a 7-point Likert-type scale ranging from 1 (completely disagree) to 7 (completely agree). An elevated score on this scale signified a higher level of life satisfaction. The McDonald’s omega coefficient of this study was 0.91 [CIs 95% 0.90; 0.91].

### 2.4. Statistical Analysis

Data analysis was conducted utilizing SPSS version 26.0, JASP version 0.18, and Mplus version 7.0. Firstly, confirmatory factor analysis (CFA) was performed on the total sample (*n* = 1446) using Mplus version 7.0. The Shapiro–Wilk normal test was performed on the C-BIFIS, and the data of each item showed a non-normal distribution (*p* < 0.001). Therefore, the Robust Maximum Likelihood (MLR) estimator was chosen. The following metrics were used to test model fit: Satorra–Bentler χ^2^ (S-Bχ^2^), Comparative Fit Index (CFI), Tucker–Lewis Index (TLI), Root Mean Square Error of Approximation (RMSEA), and Standardized Root Mean Square Residual (SRMR). Guided by the criteria established by Hu and Benlter [47], a well-fitting model was denoted by CFI/TLI values of 0.95 or above, an SRMR of no more than 0.08, and an RMSEA of no more than 0.06. Acceptable model fit was suggested by CFI/TLI values in the range of 0.90 to 0.94, SRMR values between 0.09 and 0.10, and RMSEA values ranging from 0.07 to 0.10.

Secondly, the measurement invariance of the C-BIFIS across gender was conducted using Mplus version 7.0. Following the analysis procedure of Morin et al. [48], tests of measurement invariance of the second-order factor structure across genders were conducted in three stages. In the first stage, the second-order CFA model of the C-BIFIS was tested separately for the male and female samples. In the second stage, the measurement invariance of the first-order factor model was tested through six invariance models: configural invariance (identical factor structure), weak invariance (equal factor loadings), strong invariance (equal item intercepts), strict variance (equal residual variances), variances–covariance invariance (equal factor variances and covariance), and latent mean invariance (equal latent means) [49]. If the strict invariance model for the first-order factors was supported, then the invariance of the second-order structure was verified in the following orders: (a) configural invariance; (b) invariance of second-order loadings (weak invariance); (c) invariance of second-order intercepts (strong invariance); (d) invariance of second-order disturbances (strict invariance); (e) invariance of the second-order latent variances–covariances; and (f) invariance of the second-order latent means [48,50]. Changes in the CFI, RMSEA, and SRMR are typically used to assess the cut-off values for measurement invariance [51,52]. According to Cheung et al. [51], measurement invariance was supported if the change in the CFI (ΔCFI) of the two equivalent models was 0.01 or less. Chen [51] provided a more detailed criterion. For the model to demonstrate invariant factor loadings, the change in the CFI (ΔCFI) should be less than 0.010, and the changes in the SRMR (ΔSRMR) or RMSEA (ΔRMSEA) should be less than 0.030 or 0.015, respectively. The presence of invariant intercepts or residuals was indicated if the ΔCFI was less than 0.010, the ΔSRMR was below 0.010, or the ΔRMSEA was less than 0.015 [52]. In accordance with the criteria of Morin et al. [48], the standards for variance–covariance and latent mean invariances were consistent with those for intercepts invariance.

Subsequently, bivariate correlations were used for test–retest reliability and convergent validity analyses. Hierarchical regression analysis was conducted to examine the incremental validity of the C-BIFIS in predicting intuitive eating and life satisfaction. Specifically, covariates (gender, age, and BMI) were entered in the first level, and body satisfaction, body appreciation, and unidimensional body image flexibility were placed in the second level. Two higher-order factors of the BIFIS—flexibility and inflexibility—were entered in the third level.

Finally, to assess internal consistency reliability, McDonald’s omega (ω) coefficients were calculated by the JASP 0.18. To evaluate the statistical significance of the differences between correlation coefficients, Fisher’s *r*-to-*z* transformation technique was applied [53].

## 3. Results

### 3.1. Confirmatory Factor Analysis

The results from the confirmatory factor analysis indicated that the hierarchical model, comprising two higher-order factors and eight lower-order factors, demonstrated an acceptable model fit (S-B*χ*^2^ = 1298.00, *df* = 243, CFI = 0.91, TLI = 0.90, RMSEA = 0.06, SRMR = 0.07). This outcome suggested that the model aligned well with the data. The factor loadings for all items, depicted in Figure 1, varied between 0.42 and 0.88, reflecting adequate standardized values.

### 3.2. Measurement Invariance and Gender Differences

As shown in Table 1, the indicators of the second-order CFA model fitted well for both males and females, indicating that further testing of measurement invariance across genders could be conducted. Then, the measurement invariance of the first-order factor model across genders was examined. Firstly, the configural invariance model demonstrated an adequate fit (Table 1), indicating that the latent structures of the lower-order factors are equivalent across males and females. Subsequently, the weak, strong, strict, variances–covariance, and latent mean invariance models were sequentially established to evaluate incremental fit differences. The transition from the configural to the weak invariance model revealed negligible variations in the CFI, SRMR, and RMSEA (∆CFI = −0.002, ∆SRMR = 0.003, ∆RMSEA = 0.000), all of which were within acceptable limits, thereby confirming the equivalent factor loadings across genders. Following this, the strong invariance model was compared to the weak invariance model, with minimal changes in the fit indices (∆CFI = −0.002, ∆SRMR = 0.001, ∆RMSEA = 0.000), indicating intercept equivalence. Next, the strict invariance model, compared to the strong invariance model, showed trivial deviations in the CFI, SRMR, and RMSEA (∆CFI = 0.000, ∆SRMR = 0.003, ∆RMSEA = −0.001), all falling beneath the established criteria, which supported the equivalence of residuals between genders. As shown in Table 1, variances–covariance invariance and latent mean invariance models were also supported. Consequently, the validation of these six invariance models confirmed that the eight lower-order factors of the C-BIFIS maintained strong measurement invariance across male and female respondents.

Based on the results of the measurement invariance of the first-order model, the measurement invariance of the second-order model across gender was further tested in six steps. The results showed that the variation in parameter values between all invariance models and the previous invariance model of the second-order factors (i.e., weak vs. configural, strong vs. weak, strict vs. strong, variances vs. strict, and latent mean vs. variances–covariance) was consistent with the cut-off values for which model equivalence was established (∆CFI ≤ 0.01, ∆RMSEA ≤ 0.015). This indicated that all measurement invariant models of the second-order factors were supported. In particular, the finding that the latent means of the second-order model were invariant across gender implied that there were no latent mean differences between males and females on the two higher-order factors.

### 3.3. Reliability Analysis

As presented in Table 2, the McDonald’s omega coefficient for the construct of body image flexibility was robust at 0.89, while for body image inflexibility, the value was slightly lower at 0.88. Additionally, the McDonald’s omega coefficients of the eight lower-order factors ranged from 0.69 to 0.90. In alignment with Brichacek et al. [33], only the higher-order factors were subsequently analyzed.

The test–retest reliability for body image flexibility was 0.67, and for body image inflexibility, it was 0.57 (*p*s < 0.01). Moreover, the paired-sample *t*-test indicated no significant divergence in body image flexibility scores between the initial assessment and the one-month follow-up, *t* (98) = 0.15, *p* > 0.05. Likewise, the body image inflexibility scores maintained consistency and showed no temporal change, *t* (98) = −0.09, *p >* 0.05.

### 3.4. Convergent Validity

Bivariate correlation analyses revealed an inverse relationship between body image flexibility and body image inflexibility (*r* = −0.32, *p* < 0.01). Table 3 illustrates that the unidimensional construct of body image flexibility, as assessed by the BI-AAQ-5, exhibited a positive association with body image flexibility (r = −0.17, *p* < 0.01) and a more substantial negative association with body image inflexibility (*r* = −0.45, *p* < 0.01). Fisher’s transformation confirmed that the correlation between the BI-AAQ-5 scores and body image inflexibility was substantially stronger than that with body image flexibility (*z* = −15.28, *p* < 0.001). This pattern held for males (*z* = −7.44, *p* < 0.001) as well as females (*z* = −13.35, *p* < 0.001).

Body image flexibility positively correlated with body satisfaction, body appreciation, intuitive eating, and life satisfaction (*r* = 0.27~0.51, *p*s < 0.01). In contrast, body image inflexibility demonstrated a significant negative correlation with all of the above variables (*r* = −0.19~ −0.30, *p*s < 0.01). This relationship was true for both males and females (Table 3).

### 3.5. Incremental Validity

Hierarchical regression analyses revealed that the two higher-order factors explained 2.1% of the unique variance in intuitive eating, Δ*F* (2, 1434) = 18.28, *p* < 0.001, and 2.0% of the unique variance in life satisfaction, Δ*F* (2, 1434) = 18.09, *p* < 0.001. Specifically, body image flexibility significantly positively predicted both intuitive eating (*β* = 0. 17, *p* < 0.001) and life satisfaction (*β* = 0.11, *p* < 0.001). Body image inflexibility significantly predicted life satisfaction alone (*β* = −0.07, *p* < 0.01) without a significant effect on intuitive eating (*β* = −0.01, *p* > 0.05), as shown in Table 4.

## 4. Discussion

This research conducted a linguistic adaptation of the BIFIS from English to Chinese and examined its hierarchical factor structure and measurement invariance across genders. The psychometric evaluation, encompassing assessments of reliability, convergent validity, and incremental validity, demonstrated that the Chinese version of the BIFIS possesses robust metric properties among Chinese college students.

The findings from the confirmatory factor analysis demonstrated that the second-order structure of the C-BIFIS fit well. The C-BIFIS is consistent with the structure and content of the original scale, indicating that the BIFIS has a stable factor structure among Chinese college students. Moreover, the examination of cross-gender measurement invariance confirmed that the second-order factor structure of the C-BIFIS supported all types of invariances: configural, weak, strong, strict, covariance, and latent mean, suggesting that the instrument is invariant across genders and can be utilized uniformly across male and female college student populations. In addition, the latent means of the lower- and higher-order factors were equivalent across genders. This implied that there was no difference between men and women in terms of body image flexibility and inflexibility. In line with these findings, Brichacek et al. [33] found that there was no gender difference in body image flexibility when utilizing a multidimensional assessment tool (i.e., the BIFIS). However, in terms of body image inflexibility, the current research found no difference between male and female participants, a result that challenges earlier findings. For instance, Brichacek et al. [33] reported that females exhibited notably higher levels of body image inflexibility compared to males, which was consistent with the findings using a unidimensional measure (i.e., the BI-AAQ) [24]. In contrast, Wendell et al. [4] and Dutta et al. [54] discovered no significant gender distinctions in body image inflexibility when employing a unidimensional measure (i.e., the BI-AAQ) [24,55]. It is plausible that the majority of studies have indicated a reduced degree of body image flexibility among females compared to males [2]. This could be attributed to the heightened sociocultural pressures on appearance that females typically face, prompting them to adopt more avoidance-oriented coping mechanisms in response to body image challenges [3]. Nevertheless, Rogers et al. [2] also highlighted that a limited number of studies had directly examined gender differences in body image flexibility. In previous studies, male participants were often underrepresented in study samples, which might have led to the inconsistent findings mentioned earlier. The number of male participants in the current study was also relatively small. As such, it is suggested that the number of male participants be increased in future studies, ideally by including a more varied group of males to enable a more comprehensive examination of gender differences.

The reliability analysis indicated that the C-BIFIS demonstrates good internal consistency and stability over time, consistent with the original scale results [33]. More precisely, the McDonald’s omega coefficient (ω) of the two higher-order and eight lower-order factors of the C-BIFIS ranged from 0.69 to 0.90, indicating that the C-BIFIS has high internal consistency reliability. Additionally, the current study’s test–retest reliabilities of body image flexibility and inflexibility were 0.67 and 0.57, respectively. Furthermore, the lack of significant variation in scores for body image flexibility and inflexibility over one month indicates that the C-BIFIS maintains its stability over time, which is in accordance with the results reported by Brichacek et al. [33].

Subsequent correlational analyses uncovered a significant and moderate negative association between body image flexibility and inflexibility, indicating that these two concepts are not merely opposing poles of a single spectrum but distinct and interrelated constructs. The unidimensional measure of body image flexibility via the BI-AAQ-5 demonstrated a stronger correlation with body image inflexibility than with body image flexibility itself, reinforcing the notion that body image flexibility stands apart from inflexibility. Although commonly used to measure body image flexibility, the BI-AAQ actually measured experiential avoidance (as opposed to acceptance), which was a component of body image inflexibility. The results of this study also suggest that the BI-AAQ may be more appropriate for measuring body image inflexibility or its experiential avoidance component. As body image flexibility and inflexibility were conceptually distinct, future research should be careful to differentiate the use of appropriate measurement instruments according to different research purposes. Moreover, the convergent validity assessment revealed a substantial positive correlation between body image flexibility and various positive outcomes such as body satisfaction, body appreciation, intuitive eating, and overall life satisfaction. Conversely, body image inflexibility was found to be significantly negatively correlated with these same variables. These results align with the results of Brichacek et al. [33], suggesting that the C-BIFIS also exhibits strong convergent validity.

Finally, hierarchical regression analyses revealed that both body image flexibility and inflexibility accounted for 2.1% and 2.0% increases in intuitive eating and life satisfaction, respectively, after controlling for the contributions of demographic variables, body satisfaction, body appreciation, and unidimensional body image flexibility. Notably, body image flexibility significantly predicted life satisfaction and intuitive eating, while body image inflexibility only predicted life satisfaction. While these results are not entirely in line with those of Brichacek et al. [33], they echo findings from prior research utilizing the BI-AAQ, which did not establish a predictive link between body image flexibility and intuitive eating among adult women [16]. This suggests that body image flexibility could be more instrumental in fostering healthier eating behaviors than inflexibility. It also points to a distinct structural nature of body image flexibility compared to inflexibility. These findings have implications for future empirical studies or intervention practice, as different measures may be selected for different purposes [33].

### 4.1. Implications

The current study has several theoretical and practical values. Firstly, it stands as the pioneering effort to assess the psychometric properties of the Body Image Flexibility and Inflexibility Scale (BIFIS) within a Chinese university student population, demonstrating its cross-cultural applicability.

Secondly, this study furnishes compelling evidence for the delineation of the distinct constructs of body image flexibility and inflexibility. In alignment with the original scale, the BI-AAQ-5, a unidimensional measure of body image flexibility, displayed a stronger correlation with body image inflexibility over flexibility, highlighting the distinctiveness of these constructs. Furthermore, this study suggests that body image flexibility and inflexibility might independently influence adaptive behaviors such as intuitive eating and life satisfaction, reinforcing the differentiation in the conceptualization and assessment of these constructs.

Lastly, this study contributes a nuanced measurement instrument for practical interventions concerning body image and eating behaviors among Chinese university students. The scale’s multidimensional structure allows for tailored approaches to meet various intervention goals, whether they aim to enhance body image flexibility or mitigate inflexibility.

### 4.2. Limitations and Future Directions

The present research has certain constraints. Firstly, the participants were university students using convenience sampling, most of whom were females. This may potentially restrict the generalizability of the findings. Future work should test the scale’s psychometric properties in other populations, especially encompassing more males, a broader spectrum of ages, and clinical samples. Another limitation is that the relevant variables selected in the present study to explore convergent validity were all positive. Body image flexibility is strongly linked to negative variables such as eating disorders and psychological problems [2,3]. Therefore, future studies should investigate the predictive effect of the C-BIFIS on negative consequences, such as dietary restraint and psychological distress. Finally, this is a cross-sectional study, so no conclusions can be drawn about causality. Future longitudinal studies should examine their predictive effects on other variables.

## 5. Conclusions

The Chinese version of the BIFIS is a second-order structure with two higher-order factors and eight lower-order factors, aligning with the structure of the original scale. Moreover, the C-BIFIS has cross-gender measurement equivalence, providing a valid measure for further comparisons of gender differences. The C-BIFIS also exhibits strong internal consistency, test–retest reliability, and convergent validity. Meanwhile, the two higher-order factors have unique incremental validity for life satisfaction and intuitive eating. These findings lay a foundation for future empirical research and clinical applications. In summary, the C-BIFIS exhibits favorable psychometric properties among Chinese college students, making it a valid instrument for studies related to body image flexibility within this young Chinese population.

## Figures and Tables

**Figure 1 behavsci-14-00910-f001:**
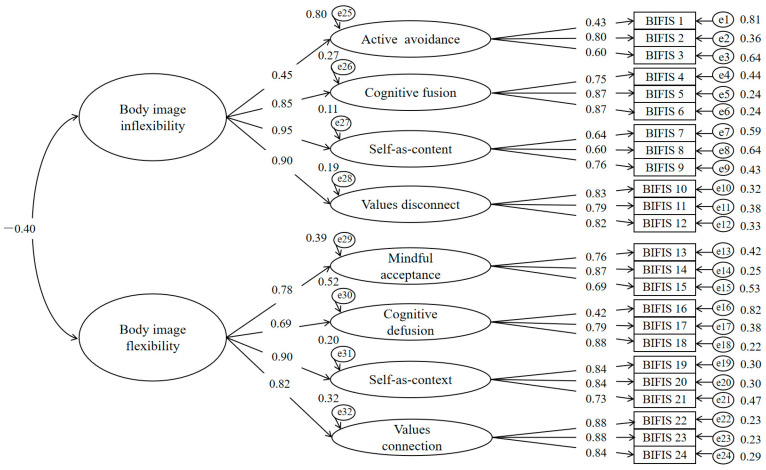
Hierarchical model of the C-BIFIS. All factor loading *is* significant at <0.001.

**Table 1 behavsci-14-00910-t001:** Results for measurement invariance test across genders (*n* = 1446).

Stages	Model	*S-Bχ^2^*	*df*	CFI	TLI	SRMR	RMSEA	∆CFI	∆SRMR	∆RMSEA
Stage 1 CFA	Males (*n* = 394)	538.035	243	0.916	0.905	0.074	0.056			
	Females (*n* = 1052)	1069.480	243	0.918	0.900	0.070	0.057			
Stage 2 first-order invariance	1—Configural invariance	1332.349	448	0.929	0.913	0.058	0.052			
	2—Weak invariance	1369.124	464	0.927	0.914	0.061	0.052	−0.002	0.003	0.000
	3—Strong invariance	1414.757	480	0.925	0.914	0.062	0.052	−0.002	0.001	0.000
	4—Strict invariance	1434.629	504	0.925	0.918	0.065	0.051	0.000	0.003	−0.001
	5—Variances–covariance invariance	1517.995	540	0.922	0.920	0.091	0.050	−0.003	0.026	−0.001
	6—Latent mean invariance	1540.601	548	0.920	0.920	0.094	0.050	−0.001	0.003	0.000
Stage 3 second-order invariance	1—Configural invariance	1699.988	542	0.907	0.906	0.077	0.054			
	2—Weak invariance	1705.128	548	0.907	0.907	0.078	0.054	0.000	0.001	0.000
	3—Strong invariance	1726.577	554	0.906	0.906	0.078	0.054	−0.001	0.000	0.000
	4—Strict invariance	1742.046	562	0.905	0.907	0.083	0.054	−0.001	0.005	0.000
	5—Variances–covariance invariance	1767.622	565	0.904	0.906	0.100	0.054	−0.001	0.017	0.000
	6—Latent mean invariance	1769.250	567	0.904	0.906	0.102	0.054	0.000	0.002	0.000

Note. Confirmatory factor analysis; S-Bχ^2^ = Satorra–Bentler χ^2^; CFI = Comparative Fit Index; TLI = Tucker–Lewis Index; RMSEA = Root Mean Square Error of Approximation; SRMR = Standardized Root Mean Square Residual; ∆CFI = Comparative Fit Index difference; ∆SRMR = Standardized Root Mean Square Residual difference; ∆RMSEA = Root Mean Square Error of Approximation difference.

**Table 2 behavsci-14-00910-t002:** Internal consistency reliability of the C-BIFIS.

C-BIFIS Factor/Dimension	No. Items	ω	95%CI	C-BIFIS Factor/Dimension	No. Items	ω	95%CI
Body image flexibility	12	0.89	[0.88, 0.90]	Body image inflexibility	12	0.88	[0.88, 0.89]
Mindful acceptance	3	0.80	[0.77, 0.82]	Active avoidance	3	0.69	[0.65, 0.72]
Cognitive defusion	3	0.76	[0.73, 0.78]	Cognitive fusion	3	0.87	[0.86, 0.88]
Self-as-context	3	0.84	[0.83, 0.86]	Self-as-content	3	0.72	[0.70, 0.75]
Values connection	3	0.90	[0.89, 0.91]	Values disconnection	3	0.85	[0.84, 0.87]

Note. BIFIS = Body Image Flexibility and Inflexibility Scale. ω = McDonald’s omega; 95%, CI = 95% Confidence Interval.

**Table 3 behavsci-14-00910-t003:** Correlations between C-BIFIS and relevant variables (*n* = 1446).

Variables	Body Image Flexibility			Body Image Inflexibility		
	Females	Males	Overall	Females	Males	Overall
Unidimensional BIF	0.22 **	0.08	0.17 **	−0.43 **	−0.47 **	−0.45 **
Body satisfaction	0.33 **	0.25 **	0.30 **	−0.30 **	−0.31 **	−0.30 **
Body appreciation	0.53 **	0.48 **	0.51 **	−0.35 **	−0.29 **	−0.33 **
Intuitive eating	0.27 **	0.29 **	0.27 **	−0.19 **	−0.19 **	−0.19 **
Life satisfaction	0.40 **	0.37 **	0.39 **	−0.29 **	−0.27 **	−0.28 **

Note. BIF = body image flexibility. ** *p* < 0.01.

**Table 4 behavsci-14-00910-t004:** Incremental validity of the C-BIFIS (*n* = 1446).

Step/Variable	Intuitive Eating		Life Satisfaction	
	*ΔR2*	*β*	*∆R2*	*β*
Step 1	0.04 ***		0.01	
Gender		−0.16 ***		−0.01
Age		0.05		0.07 *
BMI		−0.16 ***		0.01
Step2	0.13 ***		0.40 ***	
Gender		−0.14 ***		−0.01
Age		0.03		0.03
BMI		−0.08 **		0.08 **
Body satisfaction		0.13 ***		0.33 ***
Body appreciation		0.15 ***		0.40 ***
Unidimensional BIF		0.19 ***		−0.03
Step3	0.02 ***		0.02 ***	
Gender		−0.14 ***		−0.02
Age		0.02		0.03
BMI		−0.08 **		0.06 **
Body satisfaction		0.13 ***		0.32 ***
Body appreciation		0.07 *		0.34 ***
Unidimensional BIF		0.19 ***		−0.06 *
BIFIS-flexibility		0.17 ***		0.11 ***
BIFIS-inflexibility		−0.01		−0.07 **

Note. BMI = Body Mass Index; BIF = body image flexibility; BIFIS = Body Image Flexibility and Inflexibility Scale. * *p* < 0.05, ** *p* < 0.01, *** *p* < 0.001.

## Data Availability

The data of this study are available from the corresponding author upon reasonable request.

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
