# Peer review of "Validation of the Chinese Version of the Body Image Flexibility and Inflexibility Scale among Chinese College Students"

_behavsci, 2024, doi:10.3390/bs14100910_

Round 1

Reviewer 1 Report

Comments and Suggestions for Authors

This is a very well designed study that aimed to test the reliability and validity of the BIFIS among Chinese college 18 students. The authors conducted confirmatory factor analysis, convergent validity analysis, internal consistency, and test-retest. I congratulate the authors for their methodological care and use of appropriate psychometric methods.

The introduction is well written. It explains the main constructs (i.e., body image flexibility and body image inflexibility), as well as instruments available for their assessment. Furthermore, the authors explain the advantages of the BIFIS compared to other available measures.

I am convinced that making this measure available to the Chinese population is important.

I have only minor concerns that I would like the authors to address.

1.       The scales used, with the exception of the BI-AAQ, are convergent measures and not criterion measures. Please, revise accordingly.

2.       Regarding the cross-cultural adaptation, please inform if you evaluated the verbal comprehension of the target population. If not, please, include it as a study’s limitation.

3.       I did not understand why the authors used Cronbach's alpha and McDonald's omega coefficients only in the BIFIS internal consistency analysis. Why didn't you use McDonald's omega in the internal consistency analysis of the other scales used? I suggest keeping just McDonald's Omega, given that the alpha coefficient is “the lower bound of reliability”.

4.       Regarding confirmatory factor analysis, please report the multivariate normality of the data. The robust maximum likelihood estimation (MLM) may not perform well with non-normal distribution data. If necessary, use the method weighted least squares mean and variance adjusted (WLSMV).

5.       The correlation between the BI-AAQ-5 and body image inflexibility (r = -0.45, p < 0.01) could be better explored in the discussion. This justifies important differences between the two measures in conceptual terms.

Comments on the Quality of English Language

Minor editing of English language required.

Reviewer 2 Report

Comments and Suggestions for Authors

Dear Authors,

Thank you for your manuscript. The paper is well-written, well-structured, and includes all the necessary procedures for scale validation. Congratulations on your work!

Please consider the following comments:

  1. In Section 2.1 (Participants), please provide the eligibility, inclusion, and exclusion criteria. Additionally, indicate the study area(s) of the participants. Have you considered adding BMI to the study measures for correlation or regression analyses?

  2. I suggest moving the scale translation procedures to a separate section, which could be titled "Translation and Adaptation of the BIFIS" or something similar. The "Procedures" section should focus on the study's organization, including the recruitment of participants, the process of obtaining informed consent, the time required to complete the online survey, and any other relevant information related to data collection.

  3. How was missing data treated? Please provide details.

  4. The fact that the female group was dominant in this study should be mentioned as a limitation.

  5. Please explain the abbreviations used in Tables 1 and 2 within the footnotes. Additionally, include the 95% confidence intervals for alpha and omega in Table 2.

  6. Including the translated version of the scale in the supplementary file would be beneficial for future researchers who are interested in this work.

Comments on the Quality of English Language

Minor English issues detected.

Round 2

Reviewer 2 Report

Comments and Suggestions for Authors

Dear authors,

Thank you for your careful revisions. I have no more comments.

Wish you the very best of luck in your further research!

Author Response

Thank you again for all your comments and suggestions on improving our paper.